# Effects of Climatic Factors on Diarrheal Diseases among Children below 5 Years of Age at National and Subnational Levels in Nepal: An Ecological Study

**DOI:** 10.3390/ijerph19106138

**Published:** 2022-05-18

**Authors:** Meghnath Dhimal, Dinesh Bhandari, Khem B. Karki, Srijan Lal Shrestha, Mukti Khanal, Raja Ram Pote Shrestha, Sushma Dahal, Bihungum Bista, Kristie L. Ebi, Guéladio Cissé, Amir Sapkota, David A. Groneberg

**Affiliations:** 1Nepal Health Research Council, Ramshah Path, Kathmandu 44600, Nepal; me.dinesh43@gmail.com (D.B.); khemkarki9n@gmail.com (K.B.K.); sush.dahal@gmail.com (S.D.); bistabihungum@gmail.com (B.B.); 2Department of Community Medicine, Institute of Medicine, Tribhuvan University, Kathmandu 44600, Nepal; 3Central Department of Statistics, Tribhuvan University, Kirtipur 44600, Nepal; srijan_shrestha@yahoo.com; 4Management Division, Department of Health Services, Teku, Kathmandu 44600, Nepal; mukti_khanal@yahoo.com; 5National Tuberculosis Centre, Santo Thimi, Bhaktapur 44600, Nepal; 6World Health Organization, Country Office for Nepal, Lalitpur 44600, Nepal; poteshresthar@who.int; 7School of Public Health, Georgia State University, Atlanta, GA 30303, USA; 8Center for Health and the Global Environment, University of Washington, Seattle, WA 98195, USA; krisebi@gmail.com; 9Department of Epidemiology and Public Health, Swiss TPH, 4051 Basel, Switzerland; gueladio.cisse@swisstph.ch; 10Maryland Institute for Applied Environmental Health, School of Public Health, University of Maryland, College Park, MD 20742, USA; amirsap@umd.edu; 11Institute of Occupation, Social and Environmental Medicine, Goethe University, 60323 Frankfurt am Main, Germany; groneberg@med.uni-frankfurt.de

**Keywords:** climate change, Nepal, diarrhea

## Abstract

Introduction: The incidence of diarrhea, a leading cause of morbidity and mortality in low-income countries such as Nepal, is temperature-sensitive, suggesting it could be associated with climate change. With climate change fueled increases in the mean and variability of temperature and precipitation, the incidence of water and food-borne diseases are increasing, particularly in sub-Saharan Africa and South Asia. This national-level ecological study was undertaken to provide evidence linking weather and climate with diarrhea incidence in Nepal. Method: We analyzed monthly diarrheal disease count and meteorological data from all districts, spanning 15 eco-development regions of Nepal. Meteorological data and monthly data on diarrheal disease were sourced, respectively, from the Department of Hydrology and Meteorology and Health Management Information System (HMIS) of the Government of Nepal for the period from 2002 to 2014. Time-series log-linear regression models assessed the relationship between maximum temperature, minimum temperature, rainfall, relative humidity, and diarrhea burden. Predictors with *p*-values < 0.25 were retained in the fitted models. Results: Overall, diarrheal disease incidence in Nepal significantly increased with 1 °C increase in mean temperature (4.4%; 95% CI: 3.95, 4.85) and 1 cm increase in rainfall (0.28%; 95% CI: 0.15, 0.41). Seasonal variation of diarrheal incidence was prominent at the national level (11.63% rise in diarrheal cases in summer (95% CI: 4.17, 19.61) and 14.5% decrease in spring (95% CI: −18.81, −10.02) compared to winter season). Moreover, the effects of temperature and rainfall were highest in the mountain region compared to other ecological regions of Nepal. Conclusion: Our study provides empirical evidence linking weather factors and diarrheal disease burden in Nepal. This evidence suggests that additional climate change could increase diarrheal disease incidence across the nation. Mountainous regions are more sensitive to climate variability and consequently the burden of diarrheal diseases. These findings can be utilized to allocate necessary resources and envision a weather-based early warning system for the prevention and control of diarrheal diseases in Nepal.

## 1. Introduction

Climate change is an emerging public health crisis that can impact individuals across the globe. The direct impacts of climate change associated with increases in the frequency and intensity of heatwaves, extreme precipitation events, floods, droughts, and fires receive considerable attention. Less attention has been paid to indirect impacts related to ecological disruptions such as crop failures, shifting patterns of disease vectors, and increases in the burden of diarrheal disease [1]. It is estimated that ongoing climate change will have a considerable impact on the burden of diarrheal diseases [2], a leading cause of childhood morbidity and mortality in low- and middle-income countries (LMICs) such as Nepal [3].

The most recent IPCC (AR6) report suggested South Asian countries including Nepal are particularly vulnerable to the risks of climate change [4]. Available evidence indicates that the maximum temperature in Nepal is rising at a greater rate (0.05 °C/year) than the minimum temperature (0.03 °C/year) [5]. There is considerable heterogeneity in warming trends with the Mountain region experiencing a higher rate of warming compared to the plain regions [5]. Consequently, the glaciers in the Mountain region are melting and the likelihood of Glacial Lake Outburst Floods (GLOFs) is increasing [6]. Meanwhile, regions in the middle hills are facing water scarcity due to prolonged droughts and drying of spring sources [7]. Similarly, changes in precipitation patterns are leading to floods and landslides in areas with high rainfall and droughts, and water scarcity in areas with less rainfall [8]. The impact of climate change is already being experienced across different sectors, including public health. A recent report by the Department of Health Service suggests that rising temperatures are contributing to a higher incidence of water and food-borne diseases such as diarrhea, dysentery, and typhoid [9]. This is in agreement with other studies that have reported a similar association between diarrheal disease and meteorological variables including relative humidity [10], temperature [11] and precipitation [12]. Given that diarrheal diseases rank among the top five causes of death in Nepal, there is a high likelihood that the burden of diarrheal disease could increase under future climate change scenarios [13].

Although the Government of Nepal has made a significant investment in water, sanitation, and hygiene to reduce the burden of diarrheal diseases, the number of cases of diarrheal diseases remains high, and frequent epidemics of diarrhea and cholera are reported every year [14]. Although a few studies assessed the effects of weather and climate factors on diarrheal diseases in selected districts of Nepal, there is a paucity of data at a national level. Given the evidence gap, this study aimed to estimate a national and sub-national level impact of weather and climatic factors on the incidence of diarrheal disease in Nepal using long-term climate data and diarrheal disease cases from all 75 districts of Nepal (now 77 districts with further division of two districts into four). Districts are the second level of administrative divisions in Nepal. Following the introduction of the federal system of governance in 2015, districts are grouped together to form a province (major administrative units), of which there are seven.

## 2. Materials and Methods

### 2.1. Study Design

An ecological time-series study design analyzed childhood diarrheal disease cases from all districts of Nepal collected through the health management information system within the Department of Health Service, Ministry of Health and Population, Nepal.

#### Study Sites

Nepal, a mountainous country, can be categorized into three broad categories from its standard five physio-geographic regions, namely, lowland (Terai and Siwaliks), mid-mountains and hills (middle and high mountains) and high mountains (high Himalayas) from south to north [15,16]. Because there are extreme variations in altitude, the climate type ranges from tropical/subtropical in the southern Terai to polar in the northern high mountains within a short horizontal distance of less than 200 km [17]. Moving along the altitudinal transect, the southern plain of Terai region experiences tropical savanna type of climate with dry winter and hot summers (mean annual temperature 20–28 °C). This region experiences heavy rainfall during monsoon period of June–August (mean annual rainfall of 1600–1800 mm in the west and 2500–3000 mm in the east). Similarly, in the mid-hilly region, the climate is mostly sub-tropical, but it becomes temperate above the altitude of 2000 m. Finally, the northern Himalayan region has an alpine climate with temperatures reaching below −30 °C in winter. These three ecological zones (Terai, hills and mountains) are sub-divided into 15 eco-developmental regions, based on the five developmental regions (eastern, central, western, mid-western and far western): Eastern Terai (ET), Eastern Hill (EH), Eastern Mountain (ET), Central Terai (CT), Central Hill (CT), Central Mountain (CM), Western Terai (WT), Western Hill (WH), Western Mountain (WM), Mid-Western Terai (MWT), Mid-Western Hill (MWH), Mid-Western Mountain (MWM), Far-Western Terai (FWT), Far-Western Hill (FWH), Far-Western Mountain (FWM). This study included all 77 districts within these 15 eco-development regions, as shown in Figure 1.

### 2.2. Data Sources

#### 2.2.1. Hydrometeorological Data

Hydrometeorological data were sourced from the Department of Hydrology and Meteorology (DHM), Government of Nepal. Monthly temperature (maximum and minimum) and precipitation data from 2002 to 2014 were available digitally. DHM has established several local weather stations across the country; these are not uniformly distributed across the 75 districts, hence spatial scale of the hydrometeorological data varies. Data were collected from the weather stations that had a complete record of hydrometeorological information for the given district. Hydrometeorological data from local weather stations were not population-weighted.

#### 2.2.2. Outcome Variable

The outcome variable of analysis was diarrhea incidence among the children below 5 years of age, therefore, the population required to normalize the analysis consisted of the population of under 5 years children estimated by the national census 2011. Data on diarrhea cases in children less than five years old at the district level were obtained from the Health Management Information System (HMIS), Department of Health Service (DoHS) of Ministry of Health and Population for the period from July 2002 to June 2014. Annual incidence was generated by taking annual average of the monthly diarrhea incidence data for the study period.

### 2.3. Statistical Analysis

The diarrhea disease annual incidence was computed from the complete dataset for the years 2002 to 2014. The monthly and yearly trend analyses of diarrheal disease for the last 13 years were evaluated at the 15 geographic units (5 development regions × 3 ecological regions). All 75 districts had 13 years of monthly data (144 for each district) resulting in a total of 10,799 (one month missing) unit months of data for analysis. Meteorological parameters included as predictor variables were maximum temperature and minimum temperature (°C), rainfall (mm), and relative humidity (morning and evening). Monthly average of maximum and minimum temperature was computed from observed daily maximum and minimum temperature. Missing meteorological data from different stations were replaced by data from the nearest station within a domain (eco-development region).

### 2.4. Statistical Modeling

Ecological time-series modeling used monthly data from spatially dispersed districts. Estimates of variables associated with change in monthly diarrheal cases of children less than five years old were calculated separately for each of the 15 eco-development regions and a pooled estimate represented the overall effects at the national level. We used negative binomial model (NB), also known as the Poisson–gamma model with NB2 variance function to model the association between meteorological variables and diarrheal disease. This is a generalized linear model (GLM) with log link function suitable for over-dispersed count data. The model is specified as follows:(1)$$\begin{array}{l} \log\left(\lambda_{i}\right)=\beta_{0}+\beta_{1}x_{i1}+..\beta_{k}x_{ik}+\varepsilon_{i}\\ =\beta_{0}+\sum_{i=1}^{k}\beta_{k}x_{ik}+\varepsilon_{i} \end{array}$$
where $\beta_{i}$s are the unknown parameters, $x_{ik}$ are the values of the predictor variables, $\lambda_{i}$ is the mean of the dependent variable, $\beta_{0}+\varepsilon_{i}$ is the random intercept in the model. Additionally, the model can be expressed as

where $\mu_{i}=e^{\sum_{i=1}^{k}\beta_{k}x_{ik}}$ and $e^{\beta_{0}+\varepsilon_{i}}$ is the random intercept term. In the NB2 model, the variance function allows over-dispersion is $\mu_{i}+\alpha \mu_{i}^{2}$ where *α* is a scalar parameter. Predictors with *p*-values < 0.25 were retained in the fitted models. Several model adequacy tests were used to check goodness of fit, multicollinearity, heteroscedasticity, autocorrelations, and over dispersion using SPSS version 20, IBM Corp., Armonk, NY, USA. Goodness of fit as estimated by the Omnibus test was highly significant (*p*-value < 0.0001) indicating that the statistical modeling fit well with the NB model. Multicollinearity assessed by variance inflation factors (VIFs) had values that were less than 3 for all fitted models, which verified the absence of substantial multicollinearity. There were a few outliers that were deleted when obtaining the final estimates. Examination of residual plots after deletion of outliers showed a constant pattern with randomly scattered residuals in all the models, which rules out heteroscedasticity in the fitted models. The NB model was used to overcome over dispersion in the monthly diarrheal disease data. There were some significant autocorrelations that were ignored due to the very large sample sizes of the fitted models with spatially dispersed temporal data. Further, the scatterplots of the time-series residuals were randomly distributed, without any visible patterns.

## 3. Results

### 3.1. District Level (Monthly) Climatic and Diarrhea Disease Incidence Rate

The monthly time-series plots of diarrheal disease incidence rate, rainfall, and average temperature per district are shown in Figure 2. We observed an overall increasing trend of diarrheal disease rate over the study period. The trend of monthly diarrhea incidence per district ranged from 15 cases per 1000 population to 70 cases per 1000 population. Similarly, the monthly average district-level mean temperature ranged between 12 and 28 °C. Finally, the amount of average rainfall per district remained constant over the study period, ranging between 0 cm and 58 cm. As shown in Figure 2, the amount of rainfall was higher during the summer months, which represents the monsoon season in Nepal.

### 3.2. Diarrheal Incidence in Under-Five Children by Eco-Development Regions

The under-five diarrheal disease incidence rate increased in all the 15 eco-development regions over the last 13 years, from July 2002 to June 2014 (Figure 3). The thick red line shows the national incidence of diarrheal diseases (Figure 3). The average annual under-five diarrheal disease incidence for the study period was estimated to be 334 per 1000 population. The average incidence of diarrheal disease incidence increased considerably over the years, but not consistently. For instance, the average diarrheal disease incidence was 187 per 1000 in 2003, then decreased to 73 per 1000 in 2005, and increased to 546 in 2012. Likewise, we observed variation in the diarrheal disease incidence rate each year in eco-development regions. For example, the lowest incidence rate was 71 per 1000 in 2003 in the FWT region, while the highest incidence rate of 410 per 1000 in the same year was observed in the MWM region. The year 2012 marked the highest average incidence of diarrheal disease (Figure 3), with a national incidence rate of 572 per 1000 population, with considerable variability between FWH (307 per 1000 population) and MWM (939 per 1000 population).

### 3.3. Effects of Temperature

The effects of temperature on the occurrence of diarrheal disease cases were assessed separately for mean, maximum, and minimum temperatures. Overall, a 1 °C increase in mean temperature was associated with a 4.4% (95% CI: 3.95, 4.85) increase in the risk of diarrheal disease (Figure 4). The increases in risk varied by eco-development region, ranging from 0.85% for the CT region to 5.05% for the WM region. When the analyses were stratified by region, increases in diarrheal disease risk associated with a 1°C increase in mean temperature ranged from 1.46% in the Terai region to 3.42% in the mountain region. Our analyses indicate that people residing in the mountain region are more susceptible to diarrheal incidence because of higher mean temperatures compared with the other eco-belts of Nepal. Similarly, a 1 °C increase in maximum temperature was associated with an overall 3.87% (95% CI: 3.44, 4.31) increase in the risk of diarrheal diseases. The increase in the risk of diarrhea for change in maximum temperature also varied substantially between regions and ranged between 0.74 and 5.22%, with the lowest increase in the western hill region and the highest in the mid-western Terai. Regions with relatively lower increases in diarrheal disease cases were WH, WT and CT (less than a 1.3% increase per 1 °C increase in maximum temperature). Domains with higher effects were CH, FWT and MWT (more than a 4.2% increase) (Appendix A). Finally, a 1 °C increase in minimum temperature was associated with an overall 3.79% (95% CI: 3.39, 4.19) increase in the risk of diarrheal diseases in Nepal during the study period. The percent rise in diarrheal disease cases per 1 °C increase in minimum temperature also varied substantially between domains and ranged between 1.31 and 7.87%, with a minimum increase detected in the ET region and the highest in WM. Regions with relatively lower increases in diarrhea were ET, WT and FWM (less than 1.8% increase in diarrheal cases per 1 °C increase in minimum temperature). Regions with the highest effects were CH, MWT and WM (more than 3.8% increase for the same) (Appendix A).

### 3.4. Effects of Rainfall

The overall effect of a 1 cm increase in rainfall led to a 0.28% (95% CI: 0.15, 0.41) increase in the risk of diarrheal disease incidence at the national level (Figure 5). The percent rise in diarrheal disease rate varied across the eco-development regions. An increase in 1 cm rainfall was associated with a 0.37 to 0.80% increase in diarrheal cases with a minimum increase detected in the H region and the highest found in MWH.

### 3.5. Effects of Relative Humidity

Our analysis demonstrated that relative humidity is not a dominant predictor compared with temperature or rainfall in diarrheal cases (Appendix A).

### 3.6. Seasonal Effects

We estimated the risk of diarrhea incidence during the summer and spring seasons, with reference to the winter season. We did not include seasonality in the final model. We found substantially higher risks of diarrhea incidence during summer in ten out of the fifteen eco-development regions with a 22.94% (WT) to 64.94% (FWM) rise in diarrheal cases, compared to the winter. The percent increase in summer was highest in the hill region (35.22%) and lowest in the mountain region (25.7%). The overall effect was an 11.63% (95% CI: 4.17, 19.61) rise in summer with reference to the winter season (Figure 6).

Similarly, the spring season was negatively associated in only three eco-development regions, CH, MWT and FWT, with the smallest decrease detected in CH (15%) and the highest decrease in MWT (36.2%). The overall effect in Nepal was a 14.5% (95% CI: −18.81, −10.02) decrease in spring compared to the summer season (Appendix A). Lastly, the autumn season was associated with diarrheal disease cases in only four regions: WM, WH, MWH and FWH, with effects ranging between −13.44% (FWH) and 52.27% (WM), excluding the Terai, eastern and central domains of Nepal (Appendix A). The eco-belt seasonal effects were minimum in the mountain region (9.97%) and maximum in Terai (16.8%).

## 4. Discussion

This study is the first national-level study from Nepal to assess the impact of weather patterns on diarrhea across all administrative divisions. We identified the ecological belts that are highly vulnerable to diarrhea in the context of a changing climate. We further estimated that the risk of diarrhea is not evenly distributed across different ecological belts, with mountainous regions showing increased risk compared to Terai regions. Our findings showed that temperature and rainfall are significantly associated with an increased risk of diarrhea in Nepal. Likewise, the summer season represented a high-risk period for diarrheal diseases compared to the winter season. 

The Department of Health Services (HMIS) records the total count of diarrheal episodes but not the unique cases; hence, several episodes of diarrhea from a single case may be reported multiple times. As such, the incidence of diarrhea cases (per 1000) reported in our study is on the higher side and may not represent the true incidence at the population level. The trend analysis of diarrhea incidence over the 13 years (2002–2014) showed that the under-five diarrheal disease incidence rate in Nepal is increasing. Because diarrhea surveillance in Nepal is based on symptoms and is not pathogen-specific, it can be assumed that improved diagnostic facilities may not be responsible for increasing the incidence of diarrhea over the ten-year study period. These findings provide an evidence base for the Epidemiology and Disease Control Division of Nepal to allocate necessary resources for the control and prevention of diarrhea disease. Furthermore, these findings can be employed to design and establish an early warning system for forecasting diarrhea outbreaks with reference to changes in meteorological conditions.

At the national level, we estimated an increased risk of 4.4% in diarrheal disease cases among children under five years of age per 1 °C rise in mean temperature. This finding is similar to estimates from a systematic review and meta-analysis of ambient temperature and diarrheal diseases that reported a positive association of similar magnitude in low- middle- and high-income countries [3]. A regional study from Kathmandu, Nepal, reported an 8.1% increase in diarrhea among children under five years of age for every 1 °C increase in maximum temperature [18]. The higher risk detected in the study from Kathmandu may be explained by the different temperature index (maximum temperature) used in the study as well as the high number of cases reported from the Kathmandu district. Similar positive associations between temperature and diarrhea were reported from Latin America and Africa [19,20,21]. 

Mountains are considered the most fragile ecosystem and are home to the world’s poorest and most food-insecure populations [22,23]. In this study, we found that compared to other regions, mountain regions were more susceptible to increases in diarrhea incidence with increases in temperature and rainfall, while the Terai region had the smallest effect. Mountain regions in Nepal are popular tourist destinations among mountaineers and trekkers. Climate change is likely to increase warmer days conducive for trekking; this could result in the mountains being more crowded, leading to a higher risk of diarrheal disease outbreaks. For instance, a 1 °C increase in average temperature increased the risk of diarrhea among children <5 years by 5.05% in the WM region. Similarly, a study in northern India [24] showed a 5.6% increased risk of diarrhea among all age groups per every 10 °C increase in mean temperature. A study from Taiwan showed an increased impact of maximum temperature on morbidity associated with diarrhea among children less than 15 years and older adults, compared to the adult population [25]. A previous study in Nepal also found a higher diarrheal incidence in the mountains, followed by the hills and Terai [26]. However, this study showed a decreasing trend of diarrhea in Nepal, which may be due to the differences in the data considered (14 years of data from 1994 to 2007 for selected districts only).

Heavy rainfall accompanied by flooding is usually associated with outbreaks of water-borne diseases, especially in an LMIC country such as Nepal. Increases in rainfall may lead to an increase in water-borne diseases by enhancing the fecal–oral route of exposure to pathogens, particularly in LMIC settings where the prevalence of open defecation is high and access to municipal drinking water is low [24,27]. This study found that a 1 cm increase in rainfall was significantly associated with a 0.40 to 0.80% increase in diarrheal disease incidence with a minimum rise detected in the central mountain region and the highest in mid-western hills. A study from Fiji also found an independent association between extremes of rainfall and increases in diarrheal disease cases among infants after controlling for the effects of the season [28]. Another study in sub-Saharan Africa showed an increase in diarrhea prevalence with a shortage of rainfall in the dry season [29].

Diarrheal diseases, particularly water-borne and food-borne diarrhea, are common during summer seasons because of easy contamination of food and water sources as well as favorable temperatures for the spread of bacteria causing diarrhea [30,31], while viral diarrhea is more common during the winter season [32]. In our study, seasonal effects were important predictors of diarrheal disease incidence in Nepal. We found an overall increase in diarrheal cases in the summer (highest increase in far western mountains) whereas there was a decline in the spring, with reference to the winter and summer seasons, respectively.

In summary, we observed a significant association between ambient temperature, rainfall, and the incidence of diarrheal disease with wide variation across eco-development regions. The effects of climatic parameters on the incidence of diarrhea were more pronounced in mountain and hill regions compared to the Terai region. Nevertheless, our findings must be interpreted with caution due to several limitations. We considered only the weather variables, although other factors such as water supply, sanitation, population growth, etc. may play an important role in predicting the occurrence of diarrhea. Including those variables in the model could provide a more precise estimate of the effects of climatic factors on diarrhea occurrence in Nepal.

## 5. Conclusions

Climate change is likely to increase diarrheal disease incidence among children less than 5 years of age in Nepal unless preventive measures, health infrastructure, economic development of Nepalese people, etc. are improved substantially to counter the effects of climate change. Regional differences could be due to differences in socio-economic status, development level, population density, and access to WASH. These reasons need to be explored to improve surveillance and minimize known risk factors, such as lack of access to health facilities and poor sanitation. Existing diarrhea control programs should be updated to explicitly incorporate climate change to reduce the current and future burden of diarrheal diseases in Nepal.

## Figures and Tables

**Figure 1 ijerph-19-06138-f001:**
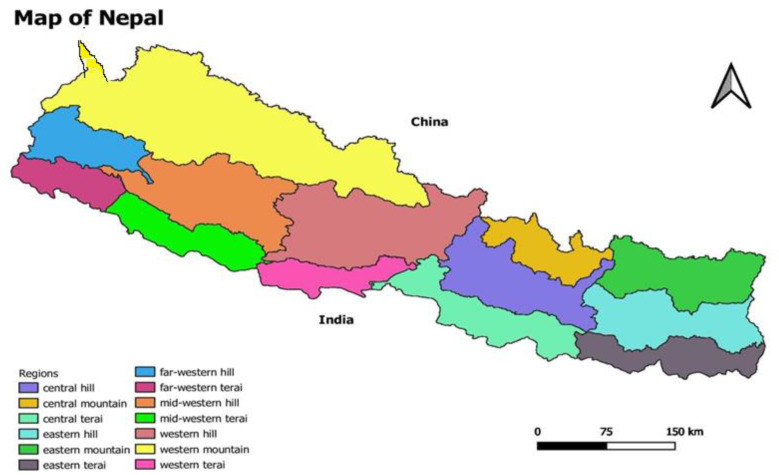
Eco-development region or cluster for study units.

**Figure 2 ijerph-19-06138-f002:**
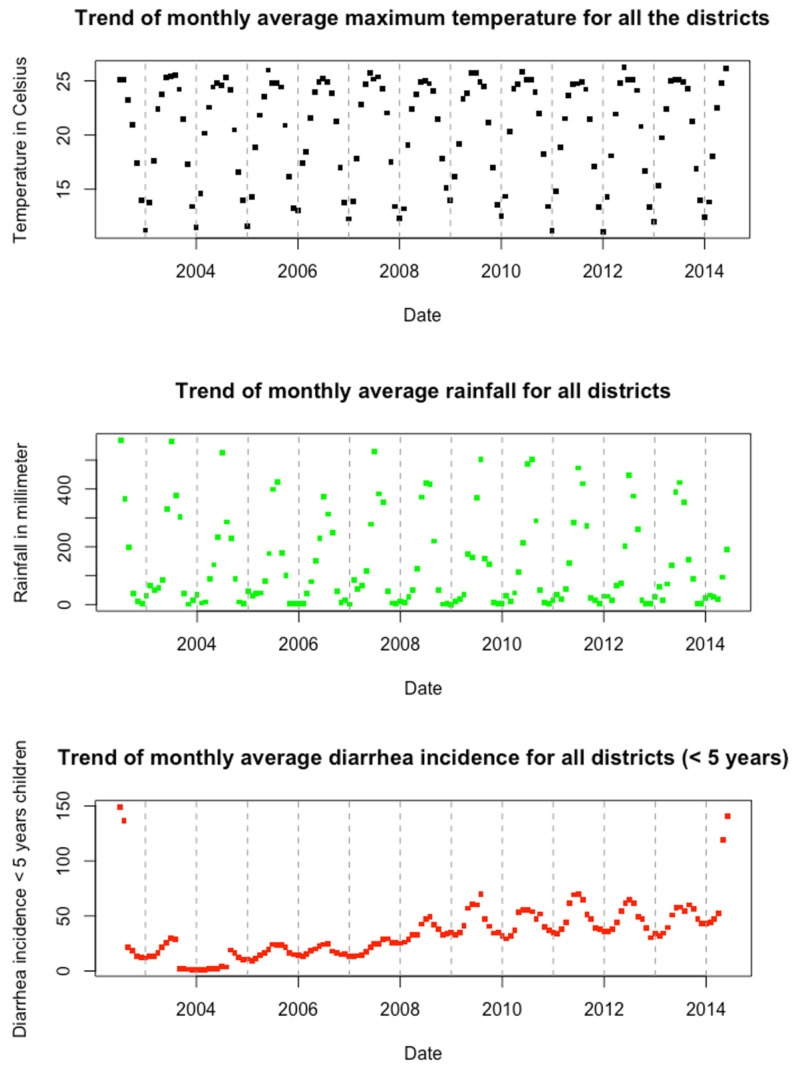
National level monthly time-series plot of climatic variables and diarrheal incidence (July 2002–June 2014) on average across all districts in Nepal.

**Figure 3 ijerph-19-06138-f003:**
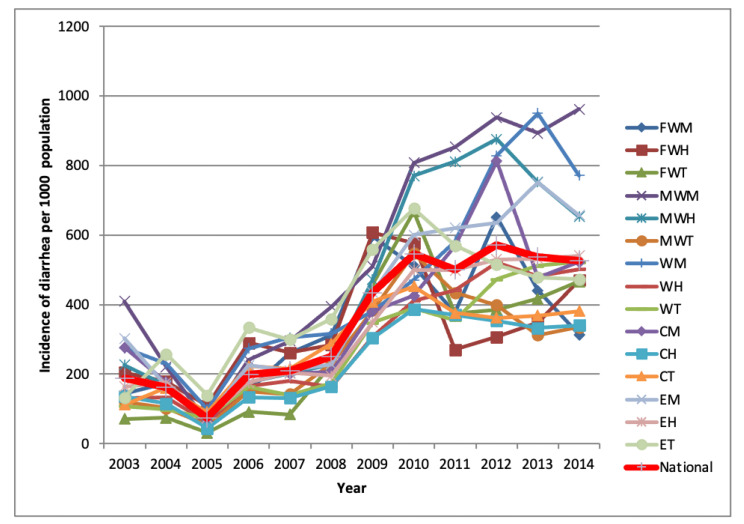
Trend of diarrheal disease by eco-development regions in Nepal **(Source: HMIS Data, 2002–2014)**.

**Figure 4 ijerph-19-06138-f004:**
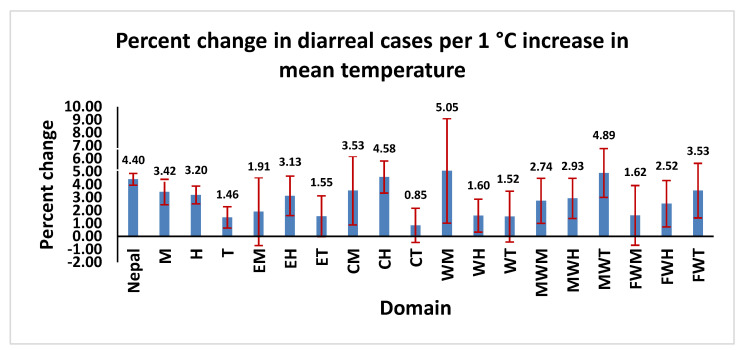
Percentage increase in childhood diarrheal cases per 1 °C increase in mean temperature per eco-development region, from July 2002 to June 2014. (Note: The projections were based on coefficients from the negative binomial time-series GLMs. The vertical lines in the figures represent the 95% confidence interval (CI)).

**Figure 5 ijerph-19-06138-f005:**
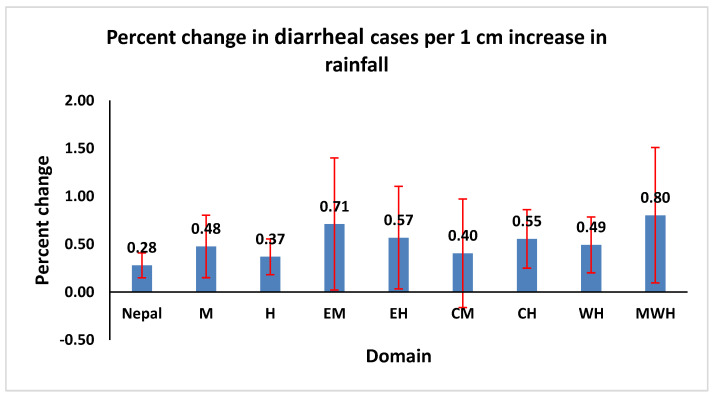
Percentage change in diarrheal cases per 1 cm increase in rainfall in eight eco-development regions, from July 2002 and June 2014. Note: data in the chart only include the domains that show significant effects. The projections were based on coefficients from negative binomial time-series GLMs.

**Figure 6 ijerph-19-06138-f006:**
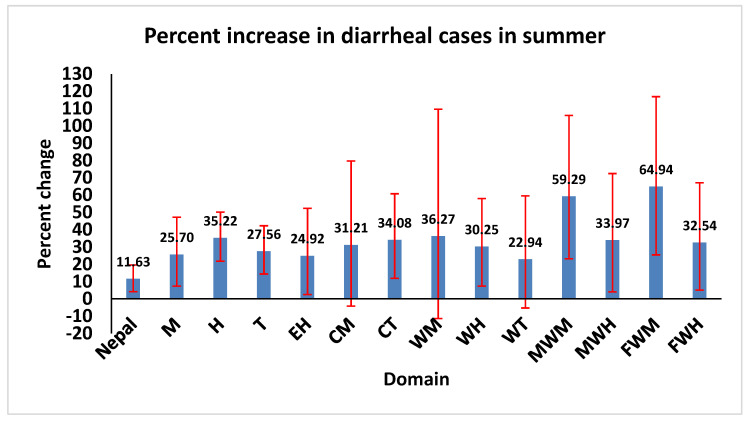
Percentage increase in diarrheal cases in summer season with reference to winter season in 13 eco-development regions, from July 2002 to June 2014. The projections were based on coefficients from negative binomial time-series GLMs.

## Data Availability

All the data supporting this study are presented in the manuscript.

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
