# Peer review of "Effects of Climatic Factors on Diarrheal Diseases among Children below 5 Years of Age at National and Subnational Levels in Nepal: An Ecological Study"

_ijerph, 2022, doi:10.3390/ijerph19106138_

Round 1

Reviewer 1 Report

Congratulations to the authors on an ambitious and timely analysis. This study addresses an important public health issue regarding the impact of meteorological conditions on diarrheal disease at a population level in a lower income country. Data sources and modeling approaches are appropriate. My concerns are with the lack of clarity of the manuscript. The methods are not described precisely enough that the analysis could be recreated.

Introduction:

This section rightly focuses on the potential for climate change to impact diarrheal disease burden in Nepal and elsewhere but could benefit from further review of the literature relating to specific meteorological variables and the direction, magnitude, and strength of their associations with diarrheal disease outcomes. Examples of previous studies that have found actionable findings using similar approaches will help justify the authors choices of study design and methodology. Some examples: PMID: 32272437; PMID: 29642611; PMID: 35024531; PMID: 28732533 (the last of these is a similarly designed analysis from another Himalayan country, and therefore highly relevant).

Materials and Methods:

Please state the study design outright here and possibly also in the article title. This was an observational, ecological study using secondary data routinely collected for other purposes.

I do not find the statistics about the latitude, longitude, surface area and elevation of Nepal very informative, relevant, or interpretable and much of this text is recycled from https://doi.org/10.3390/cli5010004. Given that this is an analysis of climate effects, please replace this text with more discussion of the variation in climate and meteorological factors across the territory of Nepal. Rather than listing the names and abbreviations of the 15 regions in the text, why not instead present them using a legend for Figure 1? This figure needs a base map so that Nepal’s location relative to neighboring countries is clear. The region labels are not legible and should instead be replaced with the acronyms in a larger font size. Please check that the color palette is colorblind friendly and consult colorbrewer2.org for general color advice for cartography.

Please clarify the spatial scale of the climate data and consider instead using the adjective “hydrometeorological” rather than “climatic” to describe them since it appears that they are time-varying (see PMID: 31229000). At what spatial scale was the exposure data matched with the outcome?

I would refer to the ecodevelopment regions as “units” rather than “clusters” or “domains” and say 10,799 unit-months of data instead of “data points”. The authors introduce the term “districts” but have not described the administrative divisions of Nepal. It should be explained how districts aggregate into regions and how these relate to the 7 provinces.

In section 2.3 the authors refer to “stations”. It was not clear from the very brief description in 2.2.1 that the data was available at the level of the weather station. Presumably that means that the locations of the stations were georeferenced in some way. How were the climate data aggregated up from the level of the weather station (point pattern) to the region (areal unit) and were they population-weighted? Unless this is clarified, I am not able to make sense of the sentence “Missing meteorological data from different stations were replaced by nearest stations within a domain (eco-development region).”

What was the outcome variable of the analysis? It appears to have been monthly, unit-level diarrhea incidence, but the population data that would be required to normalize this metric is not described. The paper title says that the analysis was restricted to children under 5 years. Where did the authors obtain the denominator estimates for this five-year age group that were used to calculate incidence? “Annual incidence” is mentioned in section 2.3, but the analysis appears to have been conducted at monthly scale.

Please give more precise definitions of the meteorological parameters and their units. Section 2.2.1 states that the estimates were monthly. Does that mean it was the maximum and minimum temperature within each month, or the monthly average of daily max and mins? Was it total rainfall volume in millimeters? Was average temperature calculated from the max and min and relative humidity from the morning and evening? I suspect that the meteorological data was actually obtained at daily resolution and was aggregated up to monthly because that was the scale at which the diarrhea case data was available, but this is not stated explicitly.

Section 2.4. appears to describe a variable selection methodology but it is not very clearly laid out. Authors should provide a complete list of “predictors”. If these were just temperature, rainfall, and humidity then it is not clear why variable selection is needed for such a small number of candidate predictors. It seems that first single variable models were fitted and then variables were retained in a final model based on their statistical significance as measured by the p-value, but this is not stated explicitly. The sentence “Predictors with p values < 0.25 are retained in the fitted models basically to capture the relevant predictors”, is vague, imprecise, and colloquial. Please state the names of the model adequacy tests that were performed.

How were seasonal effects quantified? I do not see any terms in equation 1 that might capture seasonality e.g., using Fourier/harmonic terms (see PMID: 19316455). How was unit-level clustering/autocorrelation accounted for in the model?

Results:

Figure 2: The secondary y axis is not labeled. Temperature and rainfall are plotted on the same figure but have different units. It would make more sense to present the different variables in a multi-panel figure that also includes relative humidity. The figure caption and y-axis title say that the data is “per district” but the figure legend says that it is national level. Since the data is spatially disaggregated to sub-national units, some descriptive presentation of the geographical variation in the primary exposures using choropleth maps would be helpful (see figure 1 of PMID: 34127665 as an example). The bolded caption above the figure is superfluous and actually contradicts the description in the figure title below it (does the time series start in 2002 or 2003). Please label the x axis markers with January of each year rather than the number of weeks since July 2002, which is not a helpful unit for the reader.

Please avoid evaluative language (e.g., remarkable) in the results section and keep a clear distinction between description of results and discussion. For example, paragraph 3.4 reads more like discussion and the claim that humidity affects diarrheal disease needs a citation.

Please state explicitly in the figure titles for figs 3 – 6 that these are predictions based on coefficients from negative binomial time series GLMs. It is not clear whether these results are from the stratified or pooled model.

Section 3.5. How were these seasonal effects estimated? Was it a descriptive comparison of incidence in one season compared with the rest of the year, or was there a term included in the model for season? This is not clear from the methods section.

Discussion:

To what extent could the increase in diarrhea incidence identified be due to improved reporting/healthcare access and how might this affect the interpretation of the findings?

There is some good engagement with previously published literature that puts this study in context. Discussion of potential underlying mechanisms and conceptual frameworks for interpreting the findings is lacking (e.g., the concentration/dilution hypothesis PMID: 33284047; PMID: 27058059: the enteropathogen survival/dispersal hypothesis PMID: 35024531). Further limitations that need to be addressed include: aggregating to political units is arbitrary from the point of view of transmission; associations are treated as linear; Seasonality is treated as binary; the areal unit problem; focus on all-cause diarrhea outcomes may mask conflicting pathogen-specific effects.

Author Response

We thank the reviewer for the insight full comments. We have addressed all the comments to improve the quality of the manuscript. Below is the point by point response to the comments and suggestions from the reviewer.

  1. Congratulations to the authors on an ambitious and timely analysis. This study addresses an important public health issue regarding the impact of meteorological conditions on diarrheal disease at a population level in a lower income country. Data sources and modeling approaches are appropriate. My concerns are with the lack of clarity of the manuscript. The methods are not described precisely enough that the analysis could be recreated.

Response: We have revised the methodology section to further clarify the methods and increase replicability of our work.

2.  Introduction:

This section rightly focuses on the potential for climate change to impact diarrheal disease burden in Nepal and elsewhere but could benefit from further review of the literature relating to specific meteorological variables and the direction, magnitude, and strength of their associations with diarrheal disease outcomes. Examples of previous studies that have found actionable findings using similar approaches will help justify the authors choices of study design and methodology. Some examples: PMID: 32272437; PMID: 29642611; PMID: 35024531; PMID: 28732533 (the last of these is a similarly designed analysis from another Himalayan country, and therefore highly relevant).

Response: We have added a paragraph in the introduction section to incorporate relevant literature from other parts of the world and have cited them in the reference section.

3. Materials and Methods:

Please state the study design outright here and possibly also in the article title. This was an observational, ecological study using secondary data routinely collected for other purposes.

Response: We have exclusively mentioned the study design in the methodology section.

4. I do not find the statistics about the latitude, longitude, surface area and elevation of Nepal very informative, relevant, or interpretable and much of this text is recycled from https://doi.org/10.3390/cli5010004. Given that this is an analysis of climate effects, please replace this text with more discussion of the variation in climate and meteorological factors across the territory of Nepal. Rather than listing the names and abbreviations of the 15 regions in the text, why not instead present them using a legend for Figure 1? This figure needs a base map so that Nepal’s location relative to neighboring countries is clear. The region labels are not legible and should instead be replaced with the acronyms in a larger font size. Please check that the color palette is colorblind friendly and consult colorbrewer2.org for general color advice for cartography.

Response: We have removed the information about latitude and longitude from the method section and replaced it with more information on climatic variation in Nepal. We have also replaced the map as per the suggestions of the reviewer.

5. Please clarify the spatial scale of the climate data and consider instead using the adjective “hydrometeorological” rather than “climatic” to describe them since it appears that they are time-varying (see PMID: 31229000). At what spatial scale was the exposure data matched with the outcome?

Response: We have replaced the word climatic with hydrometeorological and mentioned the spatial scale of the exposure data.

6. I would refer to the ecodevelopment regions as “units” rather than “clusters” or “domains” and say 10,799 unit-months of data instead of “data points”. The authors introduce the term “districts” but have not described the administrative divisions of Nepal. It should be explained how districts aggregate into regions and how these relate to the 7 provinces.

Response: We have replaced units with clusters to represent the ecodevelopment regions and subsequently replaced data points with unit months. We have also explained about districts and provinces to set background of administrative units in Nepal.

7. In section 2.3 the authors refer to “stations”. It was not clear from the very brief description in 2.2.1 that the data was available at the level of the weather station. Presumably that means that the locations of the stations were georeferenced in some way. How were the climate data aggregated up from the level of the weather station (point pattern) to the region (areal unit) and were they population-weighted? Unless this is clarified, I am not able to make sense of the sentence “Missing meteorological data from different stations were replaced by nearest stations within a domain (eco-development region).”

Response: We collected hydrometeorological data from weather stations located at each district. Department of Hydrology and Meteorology has established multiple weather stations in each district. We selected the weather station that had complete record of data for the study period. Data on hydrometeorological variables were not population weighted.

8. What was the outcome variable of the analysis? It appears to have been monthly, unit-level diarrhea incidence, but the population data that would be required to normalize this metric is not described. The paper title says that the analysis was restricted to children under 5 years. Where did the authors obtain the denominator estimates for this five-year age group that were used to calculate incidence? “Annual incidence” is mentioned in section 2.3, but the analysis appears to have been conducted at monthly scale.

Response: The outcome variable of the analysis was diarrhea incidence among the children below 5 years of age, so the population required to normalize it consisted of the population of under 5 years children estimated by the national census 2011. Annual incidence were generated by taking annual average of the monthly diarrhea incidence data.

9. Please give more precise definitions of the meteorological parameters and their units. Section 2.2.1 states that the estimates were monthly. Does that mean it was the maximum and minimum temperature within each month, or the monthly average of daily max and mins? Was it total rainfall volume in millimeters? Was average temperature calculated from the max and min and relative humidity from the morning and evening? I suspect that the meteorological data was actually obtained at daily resolution and was aggregated up to monthly because that was the scale at which the diarrhea case data was available, but this is not stated explicitly

Response: We have provided more precise information of meteorological parameters with their unit of measurement. Average temperature was calculated by computing mean of monthly maximum and minimum temperature. Daily temperature data were aggregated to calculate monthly average and subsequently monthly mean average temperature was calculated (from monthly average of minimum and maximum temperature).

10. Section 2.4. appears to describe a variable selection methodology but it is not very clearly laid out. Authors should provide a complete list of “predictors”. If these were just temperature, rainfall, and humidity then it is not clear why variable selection is needed for such a small number of candidate predictors. It seems that first single variable models were fitted and then variables were retained in a final model based on their statistical significance as measured by the p-value, but this is not stated explicitly. The sentence “Predictors with p values < 0.25 are retained in the fitted models basically to capture the relevant predictors”, is vague, imprecise, and colloquial. Please state the names of the model adequacy tests that were performed.

Response: The variable selection method and  other aspects of statistical analysis has been further clarified in the methodology section.

11. How were seasonal effects quantified? I do not see any terms in equation 1 that might capture seasonality e.g., using Fourier/harmonic terms (see PMID: 19316455). How was unit-level clustering/autocorrelation accounted for in the model?

Response: Seasonal effect was quantified by comparing the incidence of diarrhea during summer season with rest of the seasons. We did not specifically include any term (harmonics or splines) to capture seasonality.

12. Results:

Figure 2: The secondary y axis is not labeled. Temperature and rainfall are plotted on the same figure but have different units. It would make more sense to present the different variables in a multi-panel figure that also includes relative humidity. The figure caption and y-axis title say that the data is “per district” but the figure legend says that it is national level. Since the data is spatially disaggregated to sub-national units, some descriptive presentation of the geographical variation in the primary exposures using choropleth maps would be helpful (see figure 1 of PMID: 34127665 as an example). The bolded caption above the figure is superfluous and actually contradicts the description in the figure title below it (does the time series start in 2002 or 2003). Please label the x axis markers with January of each year rather than the number of weeks since July 2002, which is not a helpful unit for the reader.

Response: We have replaced the figure 2 with a new figure as suggested by the reviewer.

13. Please avoid evaluative language (e.g., remarkable) in the results section and keep a clear distinction between description of results and discussion. For example, paragraph 3.4 reads more like discussion and the claim that humidity affects diarrheal disease needs a citation.

Response: We have removed the evaluative language and replaced them with more general terms, where ever applicable. 

14. Please state explicitly in the figure titles for figs 3 – 6 that these are predictions based on coefficients from negative binomial time series GLMs. It is not clear whether these results are from the stratified or pooled model

Response: We have revised the titles for all the applicable figures.

15. Discussion:

To what extent could the increase in diarrhea incidence identified be due to improved reporting/healthcare access and how might this affect the interpretation of the findings?

Response: Diarrheal data are recorded in Health Management Information Systems of Nepal based on the symptoms rather than laboratory isolated etiology. Since HMIS conducts syndromic surveillance, advancement in laboratory diagnosis or improved reporting will not have significant effect in the interpretation of the findings.

16. There is some good engagement with previously published literature that puts this study in context. Discussion of potential underlying mechanisms and conceptual frameworks for interpreting the findings is lacking (e.g., the concentration/dilution hypothesis PMID: 33284047; PMID: 27058059: the enteropathogen survival/dispersal hypothesis PMID: 35024531). Further limitations that need to be addressed include: aggregating to political units is arbitrary from the point of view of transmission; associations are treated as linear; Seasonality is treated as binary; the areal unit problem; focus on all-cause diarrhea outcomes may mask conflicting pathogen-specific effects.

Response: We have included a paragraph in the discussion section to talk about concentration dilution hypothesis.

Reviewer 2 Report

Please see the comments in the attached documents. 

Author Response

We thank the reviewer for the insightful comments and suggestions. We have incorporated all the intext edits suggested by the reviewer. The point by point response to major comments are mentioned below: 

  1. It is important to differentiate between climate change and climate variability. Please contextualize climate change in the abstract.

Response: We have added a sentence in the abstract to contextualize climate change.

2. The map and font could be larger, it is not easy to read the names of the development regions.

Response: We have replaced the map with a new map where development regions are represented by different colors and subsequent legends.

3. The conclusion should tie in with your title again - you do not mention diarrhea disease among children below 5 years of age.

Response: We have corrected the conclusion and included the term diarrhea disease among children below 5 years of age.

Round 2

Reviewer 1 Report

The authors addressed my comments 7. and 8. in their responses to the review, but have not addressed them within the text of the article itself. If they could just insert the text of their responses into the article (and maybe include a subsection named "Outcome variable"), I think this will be ready to go.

Author Response

We thank the reviewer once again for the comments. We have now made necessary changes in the main text of the manuscript. The main changes include:

  1. Addition of two sentences in page number 4, data sources section. " Data were collected from the weather stations that had a complete record of hydrometeorological information for the given district. Hydrometeorological data from local weather stations were not population weighted"
  2. A separate section has been added under the subheading outcome variable. 

"The outcome variable of analysis was diarrhea incidence among the children below 5 years of age, so the population required to normalize it consisted of the population of under 5 years children estimated by the national census 2011. Data on diarrhea cases in children less than five years old at the district level were obtained from the Health Management Information System (HMIS), Department of Health Service (DoHS) of Ministry of Health and Population for the period July 2002 to June 2014. Annual incidence was generated by taking annual average of the monthly diarrhea incidence data for the study period."